# Saturated Fatty Acid Emulsions Open the Blood–Brain Barrier and Promote Drug Delivery in Rat Brains

**DOI:** 10.3390/pharmaceutics16020246

**Published:** 2024-02-07

**Authors:** Kyoung Su Sung, Won Ho Cho, Seung Heon Cha, Yong-Woo Kim, Seon Hee Choi, Hak Jin Kim, Mi Sook Yun

**Affiliations:** 1Department of Neurosurgery, Dong-A University Hospital, Dong-A University College of Medicine, Busan 49201, Republic of Korea; sungks@dau.ac.kr; 2Department of Medicine, The Graduate School of Medicine, Pusan National University, Busan 49241, Republic of Korea; 3Department of Neurosurgery, Pusan National University Hospital, Biomedical Institute of Pusan National University Hospital, School of Medicine, Pusan National University, Busan 49241, Republic of Korea; mdcwh@pusan.ac.kr (W.H.C.); neurocha@pusan.ac.kr (S.H.C.); 4Department of Radiology, Research Institute for Convergence of Biomedical Science and Technology, Pusan National University Yangsan Hospital, School of Medicine, Pusan National University, Yangsan 50612, Republic of Korea; kyw47914@pusan.ac.kr; 5Institute for Research and Industry Cooperation, Pusan National University, Busan 49241, Republic of Korea; choi3849@pusan.ac.kr; 6Department of Radiology, Pusan National University Hospital, Biomedical Institute of Pusan National University Hospital, School of Medicine, Pusan National University, Busan 49241, Republic of Korea; 7Division of Biostatistics, Research Institute for Convergence of Biomedical Science and Technology, Pusan National University Yangsan Hospital, Yangsan 50612, Republic of Korea; msyun@pusan.ac.kr

**Keywords:** saturated fatty acid, BBB, drug delivery, transmission electron microscopy, DESI-MS imaging

## Abstract

We performed this study to evaluate whether saturated fatty acid (SFA) emulsions affect the BBB and determine the duration of BBB opening, thereby promoting drug delivery to the brain. Butyric, valeric, caproic, enanthic, and caprylic acid emulsions were infused into the carotid artery of the rat model. We evaluated the BBB opening and drug delivery over time. The trypan blue and doxorubicin delivery studies were repeated from 30 min to 6 h. In the 1 h rats in each group, transmission electron microscopy (TEM) was performed to morphologically evaluate tight junctions, and the delivery of temozolomide was assessed by desorption electrospray ionization mass spectrometry. The ipsilateral hemisphere was positive for trypan blue staining in all the five SFA emulsion groups. In the valeric, enanthic, and caprylic acid emulsion groups, RGB ratios were significantly higher at 30 min and decreased thereafter. Doxorubicin delivery increased in all emulsion groups at all time points. Tight junctions were observed to be open in all groups. TMZ delivery was significantly higher in the ipsilateral hemisphere. In conclusion, intra-arterially infused SFA emulsions opened the BBB and promoted drug delivery within 30 min, which decreased thereafter. Therefore, SFA emulsions may aid BBB research and promote drug delivery to the brain.

## 1. Introduction

The blood–brain barrier (BBB) forms a structural and functional barrier between blood circulation and the brain parenchyma, preventing noxious substances from penetrating the brain parenchyma lining the blood vessel. The BBB acts as a strong barrier and a significant obstacle to drug delivery to the brain under normal conditions. The delivery of most currently available therapeutic agents is hindered by the BBB; hence, new strategies must be developed to overcome this issue. Enhanced drug transport across the BBB can aid in the treatment of brain tumors, cancer metastases to the brain, and other brain disorders [1,2].

Saturated fatty acids (SFAs) strongly affect epithelial delivery. The mechanism by which SFAs promote cutaneous permeability involves the disruption of lipids that fill the extracellular spaces of the stratum corneum, the outermost layer of the skin [3]. Therefore, SFAs are used extensively in cosmetic products. In addition, carotid arterial perfusion with sodium caprate (C10) transiently opens tight junctions of the brain endothelium [4,5]. Intraperitoneal injection of sodium octanoate (the conjugate base of caprylic acid C8) did not produce a major structural change in the BBB but facilitated brain delivery of methionine, tryptamine, and tyrosine [6]. Sodium butyrate (a sodium salt of butyric acid C4) promotes curcuminoid permeability across the BBB [7]. However, these studies have focused on short-term in vivo (within 20 min) or in vitro experiments. Meanwhile, many other SFAs may be related to BBB opening and can be explored using pharmacological strategies to improve drug delivery.

Triolein and oleic acid are commonly used to create fat embolism models, particularly in the lungs. Studies have shown that oleic acid is more toxic than triolein [8,9,10,11,12]. Free fatty acids are highly toxic to all tissues, and capillaries are particularly vulnerable [13,14]. In our previous comparative study of triolein and unsaturated fatty acids, oleic acid displayed stronger cytotoxicity and vasogenic edema in the brain [15]. Additionally, in the present study, the maximum opening time of the BBB for oleic acid was shorter than that for triolein. We wanted to understand these similarities and differences because various BBB-opening strategies have been used in various CNS diseases and patient conditions.

Doxorubicin, a popular anti-neoplastic agent, prevents growth and induces apoptosis of malignant glioma cells. It is mostly ineffective in brain tumor treatment, as it cannot cross the BBB [16]. The fluorescence characteristics of doxorubicin and its metabolites are unique and can be used to study the physiological fate of doxorubicin [17].

Temozolomide (TMZ), an imidazotetrazine derivative with broad-spectrum antitumor activity, has been used to treat malignant gliomas [18,19]. TMZ, the only chemotherapeutic agent that improves the survival rate of patients with malignant gliomas, is lipophilic and penetrates the BBB [20]. Based on the plasma/brain area under the concentration time-curve (AUC) ratio, the delivery of TMZ in the rat brain was between 35 and 39% following either oral or intravenous dosing, respectively [21]. In clinical outcome, the cerebrospinal fluid AUC values were 20% of those observed in the plasma of patients with malignant glioma [22]. Chromatographic and spectrophotometric methods such as high-performance liquid chromatography (HPLC) with ultraviolet detection and liquid chromatography coupled with tandem mass spectrometry are the most frequently used techniques for the estimation of TMZ.

Desorption electrospray ionization mass spectrometry (DESI-MS) is an efficient and highly sensitive MS ionization technique for imaging lipids and metabolites in biological tissues [23]. DESI can be used in clinical settings because it facilitates the analysis of biomolecules in the x- and y-directions by spraying charged droplets and providing chemical information as two-dimensional images [24]. Gross visualization of the increased delivery of TMZ in the animal brain may be useful for a morphologic understanding of drug delivery with the aid of the present technique.

In experimental cerebral fat embolisms induced by neutral fat (triolein) and unsaturated fatty acids, both vasogenic and cytotoxic edema appeared during the hyperacute stage [15]. Triolein emulsion infused via the intracarotid artery primarily produces vasogenic edema, resulting in reversible changes in the brain. Some saturated fatty acids are known to affect the BBB and increase drug delivery. However, the duration of BBB opening remains unknown. Therefore, in this study, we evaluated whether intra-arterially infused SFA emulsions affect the BBB, for how long they open the BBB, and whether they promote drug delivery to the brain.

## 2. Materials and Methods

### 2.1. Animal Model Establishment

The Institutional Animal Review Board of the Biomedical Research Institute approved all experimental protocols used in this study (approval no: 2022–023-A1C0, approval date: 6 May 2022). The experiments were performed using 10-week-old male Sprague–Dawley rats (Samtaco, Osan, Republic of Korea) weighing approximately 300 g after 2~3 days of acclimation. All animals were kept in a semi-specific pathogen-free environment maintained at 18–22 °C under a 12 h light/dark cycle and provided access to water and food ad libitum. Animals were anesthetized using an intramuscular injection of ketamine hydrochloride (90 mg/kg) and xylazine (10 mg/kg) and were allowed to breathe ambient air spontaneously during the procedure.

### 2.2. Experimental Protocol

First, each SFA (butyric, valeric, caproic, enanthic, and caprylic acids; Sigma-Aldrich, St Louis, MO, USA) emulsion was infused into the right common carotid artery. Particle size analysis was performed for each SFA emulsion (LS 13 320; Beckman Coulter Life Science, Indianapolis, IN, USA) to evaluate the size dimension of the emulsion. Before euthanization after SFA infusion, 1 mL trypan blue was intravenously injected into the tail vein of the rats for qualitative and semi-quantitative analyses of BBB opening. Lanthanum nitrate (0.5 mL, 5%; EMS, PA, USA), used as a TEM contrast medium, was intravenously injected for morphological evaluation of BBB opening. Doxorubicin hydrochloride (2.4 mg/kg; Boryung Pharmacy, Kyunkido, Republic of Korea) and TMZ (20 mg/kg, Sigma-Aldrich) were intravenously infused (infusion rate, 0.1 mL/s) into the rats to measure enhanced drug delivery via the BBB by SFA emulsions. The rats were euthanized using CO_2_ gas.

### 2.3. Infusion of the SFA Emulsions

Hair on the right cervical area of each rat was removed using an electric razor and a longitudinal incision was made. The right common carotid artery was isolated and the right occipital artery was ligated with a silk thread under an operating microscope (Olympus SZX7-STU2, Tokyo, Japan) to ensure that the target drug passed through the right internal carotid artery. A 24-gauge IV catheter (Angiocath; Becton Dickinson, Washington DC, USA) was inserted into the right common carotid artery. Five SFAs were used in the present study: a 1 mL syringe containing 50 µL each of butyric (C4, 13.3 mg/kg), valeric (C5, 16.66 mg/kg), caproic (C6, 19.33 mg/kg), and enanthic (C7, 21.664 mg/kg) acids, and a 10 mL syringe containing 10 mL normal saline connected to the three-way stopcock (final concentration of each four emulsion was 0.5%). Only half of this volume (25 µL) was used for the caprylic acid (C8, 11.998 mg/kg) group (the final concentration of caprylic acid emulsion was 0.25%). The SFA emulsion was prepared by mixing SFA and normal saline via vigorous to-and-fro syringe movements for 3 min. Each SFA emulsion (1.5 mL) was injected into the right common carotid artery of the rats over 30 s. After infusion of the SFA emulsion, the IV catheter was not removed from the carotid artery to maintain continuous blood flow to the artery during the experiment.

### 2.4. Study for SFA Emulsion Effect on the BBB and Drug Delivery in Time Course

The brain tissues were harvested at 30 min (*n* = 5), 1 h (*n* = 5), 2 h (*n* = 5), 4 h (*n* = 5), and 6 h (*n* = 3) after each SFA emulsion infusion. (1) The BBB opening was evaluated using trypan blue staining for qualitative and semi-quantitative analysis, and (2) quantitative analysis of doxorubicin delivery with fluorometric examination. A control study (*n* = 2) was performed with 1.5 mL saline injection via the internal carotid artery instead of SFA emulsion infusion in trypan blue staining and doxorubicin study.

#### 2.4.1. Qualitative and Semi-Quantitative Analyses of the BBB Opening with Trypan Blue Staining

For qualitative analysis of the BBB opening, whole and 1 mm slice thick harvested brain tissue sections with a rat brain cutting matrix (Kent Scientific Corp, Torrington, CT, USA) were photographed and archived as JPG files. To analyze the semiquantitative aspects of BBB opening, which was visualized as blue staining with trypan blue, the JPG files were processed using a home-made Python program language code. The color of the brain was compared 30 min, 1 h, 2 h, 4 h, and 6 h after each SFA emulsion infusion. The program extracted the red, green, and blue (RGB) values of the images from both the ipsilateral and contralateral hemispheres at specific times, and a histogram was generated to confirm the direction of the color change. Hue data were obtained from the RGB data to calculate the mean and standard deviation of each image at different times. The ratios of the RGB data in the ipsilateral to contralateral hemispheres were then obtained. Statistical analysis of differences in RGB ratios in the time course of the five SFA emulsion groups was performed using the Kruskal–Wallis test, and a *p* value less than 0.05 was considered significant. At each study time (30 min, 1 h, 2 h, 4 h, and 6 h), statistical analysis of differences in RGB ratios between each of the five SFA emulsion groups was also performed with the ANOVA test, and a *p* value less than 0.05 was considered significant.

#### 2.4.2. Quantitative Analysis of Doxorubicin Delivery with Fluorometric Examination

Doxorubicin delivery to the brain was determined via fluorometric analysis at each time point in all five groups. Fifty milligrams of the ipsilateral brain, identified via trypan blue staining, was harvested along with their contralateral counterparts. Fluorometric analysis was used to determine the concentrations of doxorubicin in the brain. A total of 50 mg of the ipsilateral brain identified by trypan blue staining was harvested, along with contralateral counterparts for controls. The tissues were then soaked in 50% ethanol in 0.3 N HCl, homogenized in a tissue blender, and refrigerated for 24 h at 4 °C. Afterward, the tissues were centrifuged at 14,000 f for 25 min. Finally, the fluorescence intensities of supernatants were measured with a fluorometer (Synergy H1, Biotek, VT, excitation/emission: 470/585 nm) at an activation wavelength of 475 nm.

Doxorubicin concentration was determined using a standard curve. Three readings were obtained for each tissue sample and averaged. To demonstrate the effect of increased vascular permeability induced by SFA emulsion on the cerebral arteries, the concentration ratios of the ipsilateral/contralateral cerebral hemispheres were calculated.

Doxorubicin delivery ratios in the ipsilateral and contralateral hemispheres were determined. Statistical analysis of the difference in doxorubicin delivery ratios over time for each SFA emulsion group was performed using the Kruskal–Wallis test, and a *p* value less than 0.05 was considered significant. At each study time (30 min, 1 h, 2 h, 4 h, and 6 h), statistical analysis of differences in RGB ratios between each of the five SFA emulsion groups was also performed with the ANOVA test, and a *p* value less than 0.05 was considered significant.

#### 2.4.3. Qualitative Analysis of TMZ Delivery via DESI-MS Imaging

For qualitative analysis of TMZ delivery under the effect of SFA on the BBB, DESI-MS images were obtained from rats in all five SFA emulsion groups harvested 1 h after each SFA emulsion infusion. DESI-MS imaging was performed on a Waters XEVO G2-XS quadrupole time-of-flight mass spectrometer (Waters, Milford, MA, USA). The DESI ion source (Waters) was mounted on a mass spectrometer and controlled using the Omni Spray software (HDI v1.6, Prosolia, Indianapolis, IN, USA). DESI imaging procedures followed Seung et al. [25]. DESI-MS imaging experiments were performed using a Waters XEVO G2-XS quadrupole time-of-flight (Q-ToF) mass spectrometer (Waters, Milford, MA, USA). The DESI ion source (Waters) was mounted on a mass spectrometer and controlled using the Omni Spray software (Prosolia, Indianapolis, IN, USA). The DESI source was initially set up using a rock spray solvent of 0.1% formic acid in acetonitrile:water (95:5, *v*/*v*) and 0.2 ng/µL leucine enkephalin (*m*/*z* 556.2771 in the ESI + mode) was added. The MS parameters were as follows: capillary voltage, 5 keV; sample cone voltage, 40 V; source temperature, 150 °C; desolvation temperature, 250 °C; desolvation flow rate, 600 L/h; cone gas flow rate, 50 L/h; gas pressure, 4.5 bar; spray voltage, 4.5 kV; spray, sprayer incidence, and collection angles, 60°, 75°, and 10°, respectively; sprayer-to-inlet and sprayer-to-sample distances, 3 mm and 1 mm, respectively; source temperature, 150 °C; and source offset, 80 V.

For the positive ion mode, the detected ion intensity of the red Sharpie marker pen (rhodamine) [M + H]^+^ at *m*/*z* 443.23 was verified. For all MS analyses, mass spectra were acquired in the *m*/*z* range of 50–700. All *m*/*z* values were extracted using a window of 0.02 Da. The square pixel size was 100 µm for the imaging MS, and the scan speeds were 100–110 µm/s. The imaging data were analyzed directly using high-definition imaging (HDI, version 1.4) with MassLynx (Waters, Milford, MA, USA, version 4.1).

### 2.5. Evaluation of Morphologic Mechanism of SFA on the BBB with Transmission Electron Microscopy (TEM)

To evaluate the morphological mechanism of SFA on the BBB, TEM examination with an intravenous injection of contrast medium (5 mL of 5% lanthanum nitrate; EMS, PA, USA) was performed in rats harvested 1 h after each SFA emulsion infusion. Some of the brain tissues that were stained blue with trypan blue were immediately cut with a blade. Five pieces with dimensions of 5 × 5 × 5 mm were obtained. These specimens were prepared for TEM evaluation as previously described [15]. TEM examinations were performed with emphasis on tight junction openings in the endothelium and neural interstitial spillage.

## 3. Results

The particle sizes in five SFA emulsions, evaluated with the particle size analyzer were as follows: butyric acid, 3.16 ± 4.06 µm, 1.91 µm; valeric acid, 3.97 ± 4.48 µm, 2.44 µm; caproic acid, 5.71 ± 7.70 µm, 2.97 µm; enanthic acid, 5.57 ± 8.17 µm, 2.49 µm; and caprylic acid, 3.77 ± 5.80 µm, 2.03 µm (mean ± standard deviation and median, respectively).

### 3.1. Study for SFA Emulsion Effect on the BBB and Drug Delivery

#### 3.1.1. Qualitative and Semi-Quantitative Analyses of the BBB Opening with Trypan Blue Staining in Time Course

In the control group, no blue staining was revealed in the ipsilateral or contralateral hemisphere in any time course. In the semi-quantitative analysis, the RGB ratios of the ipsilateral to the contralateral hemisphere of JPG files using the home-made Python code were 0.99~1.01 in the time course. Thirty minutes after the infusion of each SFA emulsion in all five groups, trypan blue staining was positive in the ipsilateral hemisphere (Figure 1). After 30 min, the blue staining tended to decrease. After 6 h, trypan blue staining was negligible in all five groups (Figure 2). In the semi-quantitative analysis of BBB opening, JPG files were analyzed using the Python code, and the RGB ratios were above 1.0 in all time courses. The ratios were the highest at 30 min and tended to decrease thereafter. At 6 h, the ratios were close to 1.00 in all five groups. In the valeric, enanthic, and caprylic acid emulsion groups, the ratios were significantly higher in 30 min and were decreased thereafter, which was statistically significant (*p* < 0.05). At 30 min, the ratios of the caproic, enanthic, and caprylic acid emulsion groups were significantly higher than those of the butyric and valeric acid groups (*p* < 0.05). In addition, the ratios in the caprylic acid emulsion group were the highest among all five SFA emulsion groups and were statistically significant (*p* < 0.05) (Figure 3).

#### 3.1.2. Quantitative Analysis of Doxorubicin Delivery with Fluorometric Examination in Time Course

In the control group, the ratios of doxorubicin delivery in the ipsilateral to the contralateral hemisphere were 1.01 in 30 min after saline infusion. In the time course of the control group, the ratios were 1.01~1.02. Doxorubicin delivery was higher in the ipsilateral hemisphere than in the contralateral hemisphere for all five SFA emulsion groups at all time points (Figure 4). After 30 min, the ratios tended to increase with increasing carbon number in the five SFA groups (butyric acid, C4; valeric acid, C5; caproic acid, C6; enanthic acid, C7; and caprylic acid, C8). In addition, the ratios in the enanthic and caprylic acid emulsion groups were significantly higher than those in the other groups (*p* < 0.05) at 30 min. In the time course of enanthic and caprylic acid emulsion groups, the ratios were highest at 30 min and decreased thereafter, which was statistically significant (*p* < 0.05). The butyric, valeric, and caproic acid emulsion groups showed no statistically significant differences over time; however, the concentration ratios were higher than 1.0, and the ratios showed a tendency to decrease over time.

#### 3.1.3. Qualitative Analysis of TMZ Delivery via DESI-MS Imaging

The signal intensity of TMZ was significantly higher in the ipsilateral hemisphere than in the contralateral hemisphere in all SFA emulsion groups 1 h after SFA emulsion infusion on DESI-MS images (Figure 5). Moreover, the signal intensity was lower in the butyric and valeric acid-treated groups than in the other groups. The signal intensity of the caprylic acid group was the highest among the five groups.

#### 3.1.4. Evaluation of Morphologic Mechanism of SFA on the BBB with TEM

Tight junctions and the neural interstitium were filled with contrast medium (lanthanum) in all five SFA emulsion groups 1 h after each SFA emulsion infusion (Figure 6). The interstitial spillage of lanthanum occurred mainly around the capillaries. The widths of the tight junctions filled with lanthanum varied from 4 to 80 nm and were nonspecific in the five SFA emulsion groups. Transcytotic vesicles containing lanthanum were not evident.

## 4. Discussion

In this study, we found that short- and medium-chain SFA emulsions opened the BBB and promoted drug delivery. BBB opening revealed blue staining on trypan blue injection, which cannot pass through the BBB to reach the brain parenchyma under normal conditions, and was most prominent at 30 min after SFA emulsion infusion in all five groups. Subsequently, BBB opening showed a tendency to decrease. Six hours after the SFA emulsion infusion, the RGB ratio of the ipsilateral to contralateral hemisphere was approximately 1.00, indicating that the BBB was closed. This BBB opening was followed by enhanced doxorubicin and TMZ delivery to the ipsilateral hemisphere in the present study. Quantitative analysis of doxorubicin delivery revealed similar results, but a similar tendency with RGB ratios (higher ratio at 30 min and decreased thereafter). Doxorubicin delivery was higher in the ipsilateral hemisphere than in the contralateral hemisphere despite no trypan blue staining at 6 h, and the reason for this discrepancy is unknown. Doxorubicin and TMZ are known substrates of the efflux transporter P-glycoprotein (P-gp), a well-characterized ABC transporter that transports various substrates across cellular membranes. Trypan blue, a vital dye, is a large molecule normally excluded from the brain by the BBB. A probable reason for the more persistent delivery of doxorubicin over trypan blue could be the remaining action of P-gp, even when the tight junctions were closed.

Neutral fat (triolein) and unsaturated fatty acid emulsions have been shown to open the BBB and increase drug delivery via intra-arterial infusion [25,26,27,28]. However, in the present study, the SFA emulsion revealed a more immediate and shorter BBB opening than the triolein emulsion noted maximum BBB opening at 2–4 h after infusion into the carotid artery. The reason for the different maximum BBB opening times among neutral fat, unsaturated fatty emulsions, and saturated fatty emulsions has not been studied, probably because of their different chemical properties, such as different molecular weights, electric characteristics, or particle sizes of the emulsion. In the present study, the mean particle size in SFA emulsions was 3–6 µm, whereas it was 15–25 µm in triolein emulsion with the same condition.

SFAs from animal and vegetable sources contain no carbon–carbon double bonds with one terminal carboxylic acid group. Most animal fats are saturated, whereas plant and fish fats are generally unsaturated. In this study, two short-chains (butyric acid [C4] and valeric acid [C5]) and three medium-chains (caproic acid [C6], acetic acid [C7], and caprylic acid [C8]) were used. Long-chain (myristic acid [C14] and palmitic acid [C16]) and very-long-chain (behenic acid [C22] and cerotic acid [C26]) emulsions became solid in the carotid artery and obstructed the vessels in our preliminary study (number of rats used = two or three for each emulsion); thus, they were not used in the present study.

In this study, the degree of BBB opening depended on the number of carbon atoms in the SFA. Semi-quantitative analysis of the RGB file of trypan blue using Python revealed statistically higher blue staining for the capric, enanthic, and caprylic acid emulsion groups than for the butyric and valeric acid groups. Similarly, higher doxorubicin delivery to the brain parenchyma resulted in a significantly higher carbon number in the SFA (in the enanthic and caprylic acid groups). Caprylic acid emulsion showed the strongest trypan blue staining and the highest concentration of doxorubicin in the ipsilateral brain parenchyma when used at half (25 µ/L) the concentration of the emulsion. Rats infused with the same concentration (50 µ/L) of caprylic acid showed strong trypan blue staining and died early within 1 h in the preliminary examination. Therefore, in the present study, half the concentration of caprylic acid was used.

BBB opening and enhanced drug delivery using intra-arterial infusion of fat emulsion produced consistent and reproducible results, although the degree differed according to the fat type. Both triolein and oleic acid were used to create fat embolism models. Non-esterified oleic acids bind to albumin and are nontoxic. However, free fatty acids are toxic to all tissues, especially capillaries [14]. However, the toxicity, intra-arterial, and intravenous effects of SFAs remain unclear. The pathogenesis of BBB opening due to fat is believed to involve a combination of mechanical and biochemical reactions [29]. The biochemical theory states that intravascular fat embolized in the lungs is converted into free fatty acids by pulmonary lipases. Free fatty acids disrupt capillary function, leading to edema, hemorrhage, and atelectasis, resulting in local inflammatory and toxic reactions. The mechanical theory states that fat occludes vessels and causes infarction. The large fat droplets mechanically disrupt the endothelium [30]. Clinical and experimental cerebral bolus fat embolisms revealed infarctions due to mechanical vascular occlusion [30]. However, the results of embolization with bolus fat or fat emulsion are quite different. In emulsions, fat particles are too small to occlude the vessel mechanically; hence, the biochemical theory is more likely to be the underlying mechanism for the results obtained. In a morphological study, a triolein emulsion opened the tight junctions of the brain endothelium, resulting in a paracellular drug delivery pathway [28]. In the present study, TEM revealed similar findings in all five SFA groups. The tight junction opens, as indicated by the filling of lanthanum contrast. Neural interstitial spillage of the contrast agent was a subsequent finding of tight junction opening due to the action of the SFA emulsion on the endothelium. Our TEM results for open tight junctions are also quite similar to those of previous studies on neutral fat emulsions on the endothelium and unsaturated fatty acid emulsions [27]. The diameter of the opened tight junction was 4–80 nm, was non-specific in the five SFA emulsion groups, and was similar in size to that of triolein (20–30 nm) [28]. However, in contrast to the triolein emulsions, transcellular vesicles were not evident in the present study. TEM results revealed a morphological mechanism by which fat emulsions open tight junctions. However, the exact molecular mechanisms underlying tight junction opening by fat emulsions remain unknown. We suspect that electric stimulation or charge change by fat electrophysiologically disrupts the tight junction bonding proteins.

In the present study, the maximum doxorubicin concentration was 2–11 times higher in the ipsilateral hemisphere than that in the contralateral hemisphere 30 min after SFA infusion. There was a tendency for higher doxorubicin delivery in the higher carbon number SFA, and the doxorubicin ratios of the enanthic and caprylic acid groups were significantly higher than those of the butyric, valeric, and caproic acid emulsion groups. Doxorubicin is easily extracted from harvested tissues and quantified fluorometrically at excitation and emission wavelengths of 470, 554, and 585 nm [31]. Doxorubicin, a popular anticancer agent, does not cross the BBB. Thus, DOX is ineffective in treating primary or metastatic brain tumors. However, an experimental permeability study of triolein emulsions revealed that doxorubicin concentrations increased in a dose-dependent manner in the brain 2 h after triolein emulsion infusion [26].

Here, the delivery of TMZ 1 h after SFA emulsion infusion was remarkably higher in the ipsilateral hemisphere than that in the contralateral hemisphere. In the caprylic acid group, the signal intensity of TMZ was highest among the five SFA groups. The results for the caprylic acid group were similar to those of trypan blue staining and doxorubicin concentration in the present study. TMZ exhibits broad-spectrum antineoplastic activity and is used to treat malignant gliomas [18,19]. With combined radiation therapy and TMZ treatment, the 2-year survival rate of patients with malignant gliomas is 24%; however, it is only 10% with radiation treatment alone [32]. However, due to resistance to TMZ treatment and the low efficacy of continuous treatment, DNA damage repair enzymes occur, resulting in the recurrence of malignancy in 60–75% of patients [33]. Higher TMZ doses may be needed for successful treatment; however, enhanced toxicity and side effects may be obstacles. To overcome these shortcomings of TMZ therapy, an efficient drug delivery system that can cross the BBB is required. In this study, qualitative analysis of TMZ concentration using DESI-MS imaging showed a significantly higher signal intensity in the ipsilateral brain than in the contralateral levels in the harvested tissues. Thus, this technique could serve as an alternative adjuvant treatment of TMZ application for malignant gliomas.

DESI-MS imaging is an emerging and useful technique that has been recently developed for metabolite imaging in tissues [34,35]. Clinically, this technique has been applied to reveal malignant tissues in brain, breast, and prostate cancers [23,36,37,38]. DESI-MS is time efficient because complex sample preparation or separation techniques are not required. Furthermore, this technique has strong spatial resolution, sensitivity, specificity, throughput, and the ability to evaluate specimens in a wide mass spectrum, from simple amino acids to drug molecules, alkaloids, terpenoids, and steroids to peptides and proteins [39,40]. The technique is also highly valuable for surface analysis, mapping, and forensic applications because it is minimally destructive and enables in situ measurements [34,39].

Not every SFA showed the same results as those in our preliminary study. For example, rats (*n* = 3) in the acetic acid emulsion infusion group died immediately after infusion. Second, long-chain SFA emulsions, namely palmitic (C16) and stearic (C18) acids, became solid when infused into the carotid arteries of rats (*n* = 2 each). Third, very long-chain SFAs, namely behenic (C22) and cerotic (C26) acids, cannot become liquids when melted with the organic solvent dimethyl sulfoxide. The melting points of behenic and cerotic acids were >80 °C, and they were unavailable for in vivo study. However, further studies are required to address these limitations.

The results of the present study showed variable BBB opening and doxorubicin delivery according to different SFA. A more prominent BBB opening indicates stronger cytotoxicity and more severe brain damage. However, these variable characteristics may be strategically useful in various brain conditions. Furthermore, the amount or density of SFA can be adjusted to avoid complications induced by overdose. This issue is not of the SFA type, but of a different pharmaceutical approach. Thus, further studies are required to better understand the intra-arterial physiological and pharmaceutical actions of fat emulsions.

The most commonly used doses of doxorubicin in rats range from 1 to 2.5 mg/kg, constituting a cumulative dose of 10–20 mg/kg [41]. Within the permissible limits of doxorubicin, the highest dosage (2.4 mg/kg) was administered in this study. However, a comparison of low and high dosages was not performed in the present study. Further research on various doses and their toxicity is necessary.

## 5. Conclusions

The SFA emulsions opened the BBB within 30 min following intra-carotid arterial infusion. BBB opening was demonstrated using trypan blue staining and TEM analysis. With sequential injection of doxorubicin and TMZ after SFA emulsion infusion, drug delivery was enhanced in the ipsilateral hemisphere, as confirmed by fluorometry and DESI-MS imaging. Among the five SFA groups, the enantiomeric and caprylic acid emulsions showed significantly greater BBB opening and higher drug delivery. Therefore, the use of SFA emulsions may be a beneficial strategy for BBB opening and promotion of drug delivery to the brain.

## Figures and Tables

**Figure 1 pharmaceutics-16-00246-f001:**
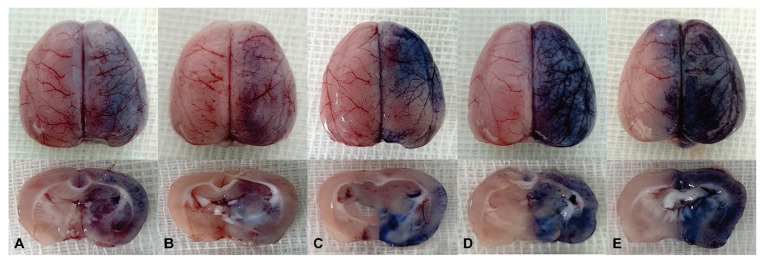
Trypan blue staining of the five saturated fatty acid (SFA) emulsion groups at 30 min infusion. Butyric (**A**), valeric (**B**), caproic (**C**), enanthic (**D**), and caprylic acid (**E**) groups. Ipsilateral cerebral hemispheres exhibited blue staining due to the blood–brain barrier opening by SFA emulsions. In enanthic (**D**) and caprylic acid (**E**) groups, the blue staining can be observed as stronger than that of the other groups.

**Figure 2 pharmaceutics-16-00246-f002:**
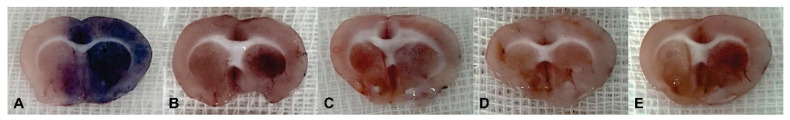
Representative figure of trypan blue staining in a rat of caprylic acid group. Blue staining is prominent in the ipsilateral hemisphere at 30 min, and decreases thereafter. (**A**), 30 min; (**B**), 2 h; (**C**), 2 h; (**D**), 4 h; (**E**), 6 h after caprylic acid emulsion infusion.

**Figure 3 pharmaceutics-16-00246-f003:**
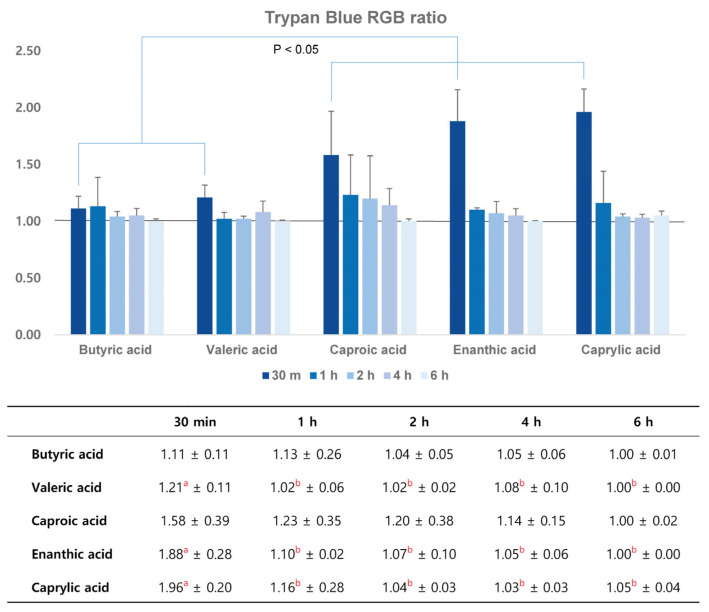
Semi-quantitative analysis of trypan blue permeability in the ipsilateral and contralateral cerebral hemispheres using the hue data. The RGB ratio of ipsilateral to contralateral hemisphere was >1.00 in all five SFA groups in all time courses. However, it had a tendency to decrease after 30 min. The ratios of caproic, enanthic, and caprylic acid groups at 30 min are significantly different from those of butyric and valeric acid groups. Among them, the ratio of caprylic acid at 30 min is the highest (*p* < 0.05). Groups represented by the same superscript letter have no difference, and there is a significant difference between groups expressed by different superscript letter.

**Figure 4 pharmaceutics-16-00246-f004:**
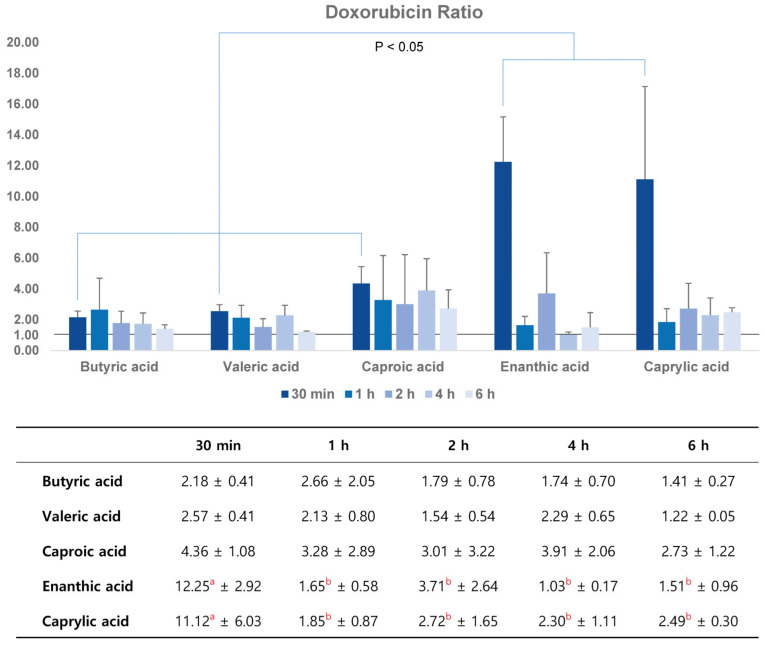
Doxorubicin concentration ratio (ipsilateral/contralateral hemisphere) in five SFA emulsion groups measured via fluorometry. Enanthic and caprylic acid groups are statistically different from butyric, valeric, and caproic acid groups at 30 min, and decrease significantly thereafter (*p* < 0.05). Groups represented by the same superscript letter have no difference, and there is a significant difference between groups expressed by different superscript letter.

**Figure 5 pharmaceutics-16-00246-f005:**
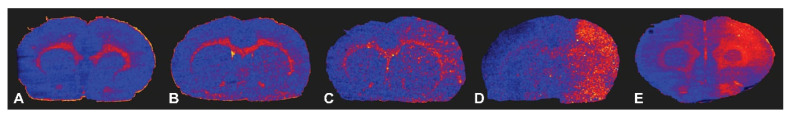
Desorption electrospray ionization-mass spectrometry (DESI-MS) images of temozolomide (TMZ) delivery to the five SFA groups at 1 h after each emulsion infusion. Minimally high signal intensity was observed in the ipsilateral hemisphere than in the contralateral hemisphere in butyric (**A**), valeric acid (**B**), and caproic acid (**C**) groups. The ipsilateral high signal intensity was marked in enanthic (**D**) and caprylic acid (**E**) groups.

**Figure 6 pharmaceutics-16-00246-f006:**
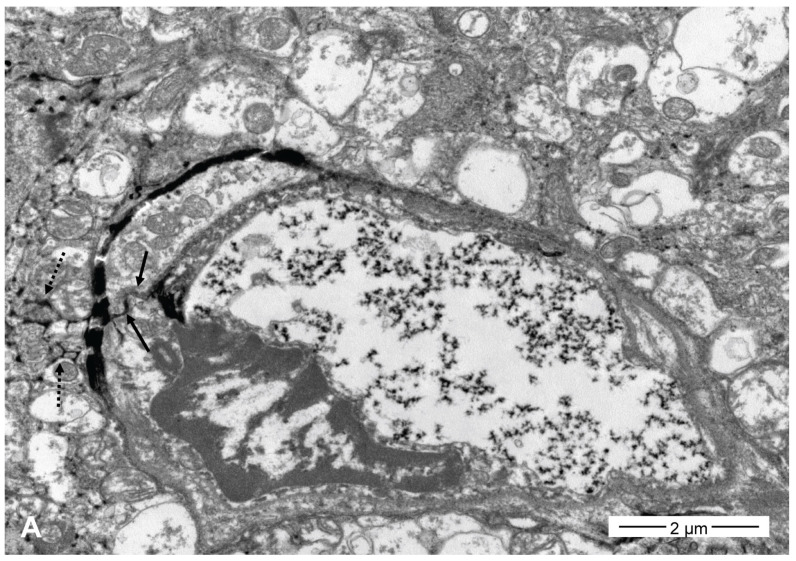
Transmission electron microscopy (TEM) of the five SFA emulsion groups at 1 h after each emulsion infusion. The tight junction (arrows) was filled, and the neural interstitium (dot arrows) was spilled with the contrast media in the butyric ((**A**), ×12,000), valeric ((**B**), ×20,000), caproic ((**C**), ×40,000), enanthic ((**D**), ×50,000), and the caprylic acid ((**E**), ×25,000) groups.

## Data Availability

Data are available upon request from the corresponding author.

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
