# Peer review of "Saturated Fatty Acid Emulsions Open the Blood–Brain Barrier and Promote Drug Delivery in Rat Brains"

_pharmaceutics, 2024, doi:10.3390/pharmaceutics16020246_

Round 1

Reviewer 1 Report

Comments and Suggestions for Authors

The paper titled, 'Saturated fatty acid emulsions open the blood-brain barrier for enhanced drug delivery' describes a novel method for drug delivery to brain. This paper conducts quantitative study using desorption electrospray ionization mass spectrometry (DESI-MS), thus demonstrating a quantitative method to study drug delivery in vivo. While this paper is a good investigative study in lab, it has hurdles to be translated clinically. 

1. The dose of doxorubicin used is way higher than in clinical settings. Did the author conduct low dosage studies? Is it possible to discuss the outcomes and challenges in the paper.  

2. Were there any toxicity studies conducted in brain and other organs of the animal post-injecting SFA. It is crucial to supplement the studies with toxicity studies, in order to provide an insight for the underlying mechanisms of SFA in BBB opening.

3. The author makes a vague statement on the mechanism of SFA in BBB opening. It is advised to propose a mechanism of action of SFA.  

This paper can be accepted for publication after major revision. 

Author Response

Authors’ Responses for #1 Reviewer’s Comments

  1. The dose of doxorubicin used in this study was much higher than that used in clinical settings. Did the authors conduct low-dose experiments? Therefore, the outcomes and challenges of this study can be discussed.   

The most commonly used doses of doxorubicin in rats range from 1 to 2.5 mg/kg, constituting a cumulative dose of 10–20 mg/kg (Ekaterina Yu Podyacheva, Ekaterina A. Kushnareva, A. Karpov, and Y. G. Toropova. Analysis of DOX-Induced Cardiomyopathy in Rats and Mice Modern views from the perspective of a pathophysiologist and clinician. Frontiers in Pharmacology. 2021;12). In the present study, we used 2.4 mg/kg doxorubicin in rats, which is approximately the highest permissible dosage. However, we did not perform low-dose studies; a study on the association between dosage and the outcome and related toxicity should be performed in the future. We have added this point as a limitation of the present study in the Discussion section as follows:

“The most commonly used doses of doxorubicin in rats range from 1 to 2.5 mg/kg, constituting a cumulative dose of 10–20 mg/kg .[46] Within the permissible limits of doxorubicin, the highest dosage (2.4 mg/kg) was administered in this study. However, a comparison of low and high dosages was not performed in the present study. Further research on various doses and their toxicity is necessary.”

  1. Were there any toxicity studies conducted in brain and other organs of the animal post-injecting SFA. It is crucial to supplement the studies with toxicity studies, in order to provide an insight for the underlying mechanisms of SFA in BBB opening.

We agree with the reviewer’s comment regarding the toxicity issues related to SFA emulsions. Unfortunately, we have not yet performed toxicity studies of SFA emulsions. In our previous reports on triolein (neutral fat) or unsaturated fatty acid emulsion infusion into the carotid artery and its toxicity study, we found no brain infarction (AJNR 2004, Investigative Radiology 2005) or hemodynamic changes (AJNR 2006) and normal proton MR spectroscopic findings (Acta Radiologica 2008). A toxicity study related to SFA emulsions will be performed in the future.

  1. The author makes a vague statement on the mechanism of SFA in BBB opening. It is advised to propose a mechanism of action of SFA.  

Thank you for this comment. In fact, the exact mechanism by which fat emulsions open the endothelium is unknown. A previous report proposed a mechanical or electrical mechanism of fat effect on the vessel or tissue; this information is included in the Discussion section of the manuscript. We suspect that electric stimulation or charge change by fat electrophysiologically disrupts the tight junction bonding proteins. We have added this information to the Discussion section of the revised manuscript.

Reviewer 2 Report

Comments and Suggestions for Authors

Current manuscript focus on the saturated fatty acid such as Butyric, valeric, caproic, enantic, and caprylic acid for delivery of Doxorubicin for brain via intra-arterially delivery.

Accept article with minor revision:

1.      Title is not clearly written. Make more attractive.

2.       Justification need on this

a)      Author mentioned drug and dye also. Both have different properties so can not predict permeation of drug based on dyes.  

b)     TEM general used for liquid sample, can you explain how TEM utilise for analysis of tissue?

3.      Add the composition of emulsion and how drug loaded

4.      How saturated fatty acid emulsion prepared ? as oil also contains saturated fatty acid. What is final concentration of Saturated fatty acid ?

5.      How doxorubicin quantified? As only mention preparing standard curve. Add more detail for quantification of doxorubicin .

6.      How brain cut into 5 × 5 mm pieces ?

7.      What is control taken for studies?

8.      Also need to add probably mechanism how fatty acid enhanced permeation of drug through bbb.

9.      Why arterial delivery choose ? why not intravenous delivery? Arterial delivery need to higher precision and expertise.

10.   Need to revised figure 1 title, if want to show protocol of animal studies then show separately. And figure 1 is considered as graphical abstract. Do needful correction for better understanding.

Author Response

Authors’ Responses for #2 Reviewer’s Comments

  1. Title is not clearly written. Make more attractive.

The title has been changed to “Saturated fatty acid emulsions open the blood-brain barrier and promote drug delivery in rat brain”

  1. Justification need on this
  2. a)       The authors also mention drugs and dyes. Both have different properties; therefore, drug permeation cannot be predicted using dyes.   

      We agree with the reviewer on this point. The drug and dye have different properties, but they both are incapable (or nearly incapable, since a small percentage of TMZ can transit) of crossing the blood-brain barrier (BBB). We wanted to determine whether different dyes or drugs that do not normally cross the BBB can enter the brain parenchyma when the BBB is opened with the aid of an SFA emulsion. We observed that materials with different properties exhibited increased drug delivery to the brain. The findings indicate that the delivery of many other drugs to the brain could be promoted by this technique. The present study revealed a little bit discrepancy of doxorubicin delivery and trypan blue staining, also. Thus we have described this issue in the Discussion section as follow: Doxorubicin and TMZ are known substrates of the efflux transporter P-glycoprotein (P-gp), a well-characterized ABC transporter that transports various substrates across cellular membranes. Trypan blue, a vital dye, is a large molecule normally excluded from the brain by the BBB. A probable reason for the more persistent delivery of doxorubicin over trypan blue could be the remaining action of P-gp, even when the tight junctions were closed.

  1. b)     TEM general used for liquid sample, can you explain how TEM utilise for analysis of tissue?

TEM is not designed to directly analyze liquid samples or samples containing liquids because examination relies on the absence of air. However, to analyze the microstructure of biological specimens, such as cells or tissues, chemicals are necessary for sample preprocessing. This preprocessing includes removing all the moisture contained inside while minimizing structural changes in cells or tissues. Successful preprocessing permits the accurate histological analysis of microstructures of biological specimens using TEM. The TEM used to examine liquid samples that is mentioned by the reviewer is probably Cryo TEM, which is actively used for the structural analysis of proteins.

In the present study, TEM was used to determine the morphological mechanism of BBB opening by the SFA emulsion. In our previous studies, TEM revealed tight junction openings with the aid of lanthanum nitrate contrast medium, which fills the junctions and spills out into the neural tissue interstitium (references 24 and 29 in the manuscript). In these studies, the diameter of the tight junction was measurable. New findings suggest that compatible sizes of nanomaterials suitable for BBB studies should be less than 30 nm.

  1. Add the composition of emulsion and how drug loaded

The SFA emulsion used in this study was a mixture of the five SFAs and normal saline. The SFA emulsion was infused, followed by trypan blue, doxorubicin, and TMZ. However, these materials have not been used as conjugate drugs.

  1. How was the SFA emulsion prepared ? because oil also contains saturated fatty acids. What is the final concentration of saturated fatty acids ?

Five SFA samples were purchased commercially (Sigma-Aldrich) as liquid preparations. Each SFA emulsion was prepared as described in the Methods section of the paper. Briefly, a 1-mL syringe containing 50 µL each of butyric (C4), valeric (C5), caproic (C6), and enanthic (C7) acids and a 10-mL syringe containing 10 mL of normal saline were connected to the three-way stopcock. Only half of this volume (25 µL) was used for the caprylic acid (C8) group. The SFA emulsion was prepared by mixing SFA and normal saline using vigorous to-and-fro syringe movements for 3 min.

The final concentration of the four SFA emulsions (butyric, valeric, caproic, and enanthic acids) was 0.5%, and of caprylic acid emulsion was 0.25%. We have added these final concentrations to the Methods section of the revised paper.

  1. How doxorubicin quantified? As only mention preparing standard curve. Add more detail for quantification of doxorubicin .

We have added the details of doxorubicin quantification using a fluorometer in the Methods section of the revised paper as follows:

“A fluorometric analysis was performed to determine the concentration of doxorubicin in the brain. The ipsilateral brain (50 mg) identified by trypan blue staining was harvested along with the contralateral counterparts as controls. The tissues were then soaked in 50% ethanol in 0.3N HCl, homogenized in a tissue blender, and refrigerated for 24 hours at 4°C. Subsequently, the tissues were centrifuged at 14,000 g for 25 min. Finally, the fluorescence intensity of the supernatant  was measured using a Synergy Hi fluorometer (BioTek Instruments, VT) at excitation and emission wavelengths of 470 and 585 nm, respectively, at an activation wavelength of 475 nm.”

  1. How brain cut into 5 × 5 mm pieces ?

      We have revised the method used to obtain the 5 × 5 mm pieces in the Methods section of the revised paper as follows:

      “Some of the brain tissues that were stained blue with trypan blue were immediately cut with a blade. Five pieces with dimensions of 5×5×5 mm were obtained. These specimens were prepared for TEM evaluation as previously described [15].”

  1. What is control taken for studies?

A control experiment was also performed. We have added the data of the control group in the Methods and Results sections of the revised paper as follows:

In the Methods section; A control study (n = 2) was performed with 1.5 mL saline injection via the internal carotid artery instead of SFA emulsion infusion in trypan blue staining and doxorubicin study.

In the Results section; In control group, no blue staining was revealed in the ipsilateral or contralateral hemisphere in all time course. In semi-quantitative analysis, the RGB ratios of the ipsilateral to the contralateral hemisphere of JPG files using the home-made Phyton code were 0.99 ~ 1.01 in time course.

.

  1. Also need to add probably mechanism how fatty acid enhanced permeation of drug through bbb.

Thank you for this comment. The exact mechanism by which fat emulsions open the endothelium is unknown. In a previous report, the effect of fat on the vessel or tissue was proposed to be mechanical or chemical, as described in the Discussion section. We suspect that electric stimulation or charge change by fat electrophysiologically disrupts the tight junction bonding proteins. We have added this information to the revised Discussion section.

  1. Why arterial delivery choose ? why not intravenous delivery? Arterial delivery need to higher precision and expertise.

Thank you for your comments. Over the past 20 years, we have often noted that the BBB remains intact upon intravenous infusion of fat emulsion. This is probably because fat emulsions are fluid, deformable, and can penetrate capillaries. After intravenous infusion, the particles pass into the lungs. Fat particles break into smaller globules as they pass through the lungs. The cycle of systemic circulation is repeated, and the globules become increasingly smaller. When they approach micron size, they are readily removed from the blood by phagocytic systems in the liver and other organs.

  1. Need to revise the title of Figure 1 if we want to show the protocol of animal studies, then show them separately. Figure 1 shows a graphical abstract. Needful corrections are required for a better understanding.

The title of Figure 1 has been changed to Graphical abstract.

Reviewer 3 Report

Comments and Suggestions for Authors

The current study explores the time course of blood-brain barrier (BBB) opening by a series of 5 short and medium chain saturated fatty acids (SFA). The same lab has over the years performed a number of similar studies using triolein or unsaturated fatty acids. While a direct comparison of those previous data with the current work is difficult, because different species (rabbits, cats, rats), different methods and doses of administration of the BBB modifying agents, and different analytical techniques were applied, the present manuscript is most similar to the study on uptake of temozolomide in rats after triolein emulsion (ref. 19). The experiments here demonstrate a short-term effect at 30 min by heptanoic and octanoic acid emulsions, but not by the shorter fatty acids. This is an interesting observation, although the studies do not provide any mechanistic explanation.

The reviewer has the following concerns with the manuscript as submitted:

Major points of critique:

1)    The rationale to test various SFAs is not well worked out. Did the authors expect significant advantages over emulsions of triolein? 

2)    When analyzing the time course of BBB effects of the 5 SFAs, it would appear logical to compare to the previously used triolein, where effects were studied at the single time point of 1h after administration. Why was that not done?

3)    The generation of stable emulsions without emulsifiers is challenging. While emulsifiers could have effects of their own and alter the effects of the emulsions, the authors do not provide a rational for not trying out biocompatible emulsifiers, e.g. lecithin. 

4)    The results of the particle size analysis indicate that the lipid droplets of all 5 emulsions cover a wide size range. In all cases there must be substantial fractions of droplets with sizes of 7-10 or more µm, which would embolize small arterioles and capillaries. Have the authors explored alternative techniques, like ultrasonication or microfiltration, to generate emulsions of more uniform, smaller droplet sizes?

5)    All the experimental data were analyzed by comparing the ipsilateral and contralateral hemispheres. Including a control group (saline injected via intracarotid artery) may be necessary to rule out the effect of any damage or BBB disruptions from the surgery or any other factors than the SFA emulsions.

6)    No details for the semi-quantitative analysis of BBB opening by RGB ratios for trypan blue are provided. Who wrote the Python code, is there any reference in the literature for this application? 

7)    Only a single dose was studied for all 5 SFA, so it is uncertain if there is a dose response relation, or what that relation would look like. The only hint is that the dose of caprylic acid had to be reduced by half compared to the other SFAs. This may indicate that the applied dose, 1.5 mL emulsion of 50 µL SFA/10 ml NaCl, for the other SFAs may also be close to the maximum tolerated dose. In this regard, it would also be preferable to express administered doses in some commonly used units. For example, in their previous paper the authors expressed triolein doses in units of mg/kg.

8)    TMZ delivery and TEM experiments were done at 1 hr after the emulsion infusion, although the highest concentrations for trypan blue and doxorubicin were measured at 30 min. Why?

9)    Doxorubicin and TMZ are the known substrates of the efflux transporter p-gp. Would this explain the discrepancy in the results between trypan blue and doxorubicin? 

Minor points:

10) In the methods section (2.2), it is mentioned that doxorubicin and TMZ doses were intravenously infused. What was the infusion rate?

11) The manuscript needs careful revision for a number of typographical or grammatical errors. For example, enanthic acid is misspelled as “enantic” acid throughout? On line 98, “after203 days” should likely read “after 2-3 days.”

12) The reviewer wonders if the stated doses of the anesthetics are simply typos? “….an intramuscular injection of ketamine hydrochloride (2.5 mg/kg) and xylazine (0.125 mg/kg) ….” on line 101. These doses are far below doses sufficient for surgical anesthesia in rats; typical doses are in the range of 80-100 mg/kg ketamine and 5-10 mg/kg xylazine. The same low doses were previously given by the authors in their triolein paper (ref. 19).

Comments on the Quality of English Language

Some minor revisions necessary, but generally acceptable language.

Author Response

#3 Reviewer’s Comments

Major points of critique:

1)    The rationale to test various SFAs is not well worked out. Did the authors expect significant advantages over emulsions of triolein? 

Triolein and oleic acid are commonly used to create fat embolism models, particularly in the lungs. Studies have shown that oleic acid is more toxic than triolein is.[8-12] Free fatty acids are highly toxic to all tissues, and capillaries are particularly vulnerable.[13,14] In our previous comparative study of triolein and unsaturated fatty acids, oleic acid displayed stronger cytotoxicity and vasogenic edema in the brain.[15] Additionally, in the present study, the maximum opening time of the BBB for oleic acid was shorter than that for triolein. We wanted to understand these similarities and differences because various BBB-opening strategies have been used in various CNS diseases and patient conditions. We have added this in the revised Introduction section.

2)    When analyzing the time course of the BBB effects of the five SFAs, it would appear logical to compare them to the previously used triolein, where the effects were studied at a single time point of 1h after administration. Why was that not done?

We agree with the reviewer’s comments. We compared the effects of trioleins in a preliminary study in the present research. However, we focused only on the SFA this time. Therefore, a comparative study will be planned in the future.

The single time point of 1 h after SFA infusion was chosen because TEM and DESI imaging studies are qualitative and not quantitative in nature. Thus, we thought that the acquisition of TEM and DESI images at 30 min or 1 h would not make any difference. A single study for the acquisition of these images 30 min after SFA emulsion infusion would be possible in terms of time.

3)     Generation of stable emulsions without emulsifiers is challenging. While emulsifiers could have effects on their own and alter the effects of the emulsions, the authors did not provide a rationale for not using biocompatible emulsifiers such as lecithin.  

We performed fat emulsion studies multiple times using several surfactants, but we did not observe any effect on BBB opening with a surfactant-coated fat emulsion. This is likely because the other materials coated with lipid particles have different mechanisms of action in the endothelium.

4)     Particle size analysis indicated that the lipid droplets of all five emulsions covered a wide size range. In all cases there must be substantial fractions of droplets with sizes of 7-10 or more µm, which would embolize small arterioles and capillaries. Have the authors explored alternative techniques, such as ultrasonication or microfiltration to generate emulsions with more uniform and smaller droplet sizes?

Thank you for this comment. Fat emboli differ from most other emboli in that they are fluid, deformable, and can penetrate capillaries. Emboli break into smaller globules as they penetrate the capillaries. After a temporary delay in the systemic capillaries, the emboli pass into the veins and return to the lungs. This cycle is repeated, and the globules become increasingly smaller until they are no longer embolic. When they approach micron size, they are readily removed from the blood by phagocytic systems in the liver and other organs. Hence, the vascular occlusion is often temporary, incomplete, or both (Sevitt S. The significance and pathology of fat embolism. Ann Clin Res 1977;9:173–180). In our previous studies using bolus (not emulsified) fat injection, we noticed infarction in the brain (2001, Investigative Radiology). However, when we used fat emulsion, there was no brain infarction (2004 AJNR, 2005 Investigative Radiology, 2006 AJNR, 2010 Acta Radiologica). We have been trying to regulate the size of the fat particle using many techniques including microfluidics.

5)    All experimental data were analyzed by comparing the ipsilateral and contralateral hemispheres. Including a control group (saline injected via the intracarotid artery) may be necessary to rule out the effect of any damage or BBB disruption from the surgery or any factors other than the SFA emulsions.

A control experiment was also performed. We have added the data of the control group in the revised Methods and Results sections as follows:

In the Methods section; A control study (n = 2) was performed with 1.5 mL saline injection via the internal carotid artery instead of SFA emulsion infusion in trypan blue staining and doxorubicin study.

In the Results section; In control group, no blue staining was revealed in the ipsilateral or contralateral hemisphere in all time course. In semi-quantitative analysis, the RGB ratios of the ipsilateral to the contralateral hemisphere of JPG files using the home-made Phyton code were 0.99 ~ 1.01 in time course.

6)    No details of the semi-quantitative analysis of BBB opening using RGB ratios for trypan blue were provided. Who wrote the Python code and is there any reference in the literature for this application? 

We have explained this issue in the Methods section as follows:

“To analyze the semiquantitative aspects of BBB opening, which was visualized as blue staining with trypan blue, the JPG files were processed using a home-made Python program language code. The color of the brain was compared 30 min, 1 h, 2 h, 4 h, and 6 h after each SFA emulsion infusion. The program extracted the red, green, and blue (RGB) values of the image from both the ipsilateral and contralateral hemispheres at specific times, and a histogram was generated to confirm the direction of the color change. Hue data were obtained from the RGB data to calculate the mean and standard deviation of each image at different times. The ratios of the RGB data in the ipsilateral to contralateral hemispheres were then obtained.”

7)    Only a single dose was studied for all five SFA; therefore, it is uncertain whether there is a dose-response relationship or what that relationship would look like. The only hint was that the dose of caprylic acid had to be reduced by half compared to the other SFAs. This may indicate that the applied dose, 1.5 mL emulsion of 50 µL SFA/10 ml NaCl, for the other SFAs may also be close to the maximum tolerated dose. In this regard, it would be preferable to express the administered doses in commonly used units. In a previous study, the authors expressed triolein doses as mg/kg.

We have changed this issue in the Methods section as follows:

Five SFAs were used in the present study: a 1-mL syringe containing 50 µL each of butyric (C4, 13.3 mg/kg), valeric (C5, 16.66 mg/kg), caproic (C6, 19.33 mg/kg), and enanthic (C7, 21.664 mg/kg) acids and a 10-mL syringe containing 10 mL normal saline connected to the three-way stopcock (final concentration of each four emulsion was 0.5%). Only half of this volume (25 µL) was used for the caprylic acid (C8, 11.998 mg/kg) group (final concentration of caprylic acid emulsion was 0.25%).

8)    TMZ delivery and TEM experiments were performed 1 h after emulsion infusion, and the highest concentrations of trypan blue and doxorubicin were measured at 30 min. Why?

The TEM and DESI imaging studies in the present study were qualitative and not quantitative in nature. We thought that the acquisition of TEM and DESI images at 30 min or 1 h would not make any difference. The images obtained after 1 h revealed opening of the tight junction on TEM and high uptake on DESI imaging, and these results were not different from our purpose of the present study.

9)    Doxorubicin and TMZ are known substrates of the efflux transporter P-gp. This could explain the discrepancy in the results between trypan blue and doxorubicin.  

We agree with the reviewer’s comments. Doxorubicin and TMZ are known substrates of the efflux transporter P-glycoprotein (P-gp), a well-characterized ABC transporter that transports various substrates across cellular membranes. Trypan blue, a vital dye, is a large molecule normally excluded from the brain by the BBB. A probable reason for the more persistent delivery of doxorubicin over trypan blue could be the remaining action of P-gp, even when the tight junctions were closed.

We have added this description in the revised Discussion section.

Minor points:

10) In the methods section (2.2), it is mentioned that doxorubicin and TMZ doses were intravenously infused. What was the infusion rate?

We have added the infusion rate of doxorubicin and TMZ as “(infusion rate, 0.1 ml/s)” in the revised Methods section.

11) The manuscript needs careful revision for a number of typographical or grammatical errors. For example, enanthic acid is misspelled as “enantic” acid throughout? On line 98, “after203 days” should likely read “after 2-3 days.”

Thank you for your comments. We have corrected the errors and had the manuscript edited by a professional English editor.

12) The reviewer wonders whether the stated doses of anesthetics are simply typographical errors. “….an intramuscular injection of ketamine hydrochloride (2.5 mg/kg) and xylazine (0.125 mg/kg) ….” on line 101. These doses are far below those sufficient for surgical anesthesia in rats; the typical doses are in the range–80-100 mg/kg ketamine and 5-10 mg/kg xylazine. The same low doses were previously reported in a triolein study (ref. 19).

We apologize for the typographical errors regarding the doses of anesthetics. We have revised the text as follows:

“Animals were anesthetized using an intramuscular injection of ketamine hydrochloride (90 mg/kg) and xylazine (10 mg/kg) and were allowed to breathe ambient air spontaneously during the procedure.”

Round 2

Reviewer 1 Report

Comments and Suggestions for Authors

The authors have addressed the comments of the reviewer satisfactorily. The paper can now be accepted for publication. 

Reviewer 2 Report

Comments and Suggestions for Authors

I am satisfied with the revisions done by the authors. It can be accepted. 

Reviewer 3 Report

Comments and Suggestions for Authors

The authors have addressed my comments.